# Cellular and Humoral Responses Follow-up for 8 Months after Vaccination with mRNA-Based Anti-SARS-CoV-2 Vaccines

**DOI:** 10.3390/biomedicines10071676

**Published:** 2022-07-12

**Authors:** Sergio Gil-Manso, Diego Carbonell, Verónica Astrid Pérez-Fernández, Rocío López-Esteban, Roberto Alonso, Patricia Muñoz, Jordi Ochando, Ignacio Sánchez-Arcilla, Jose M Bellón, Rafael Correa-Rocha, Marjorie Pion

**Affiliations:** 1Advanced Immunoregulation Group, Gregorio Marañón Health Research Institute (IiSGM), Gregorio Marañón University General Hospital, 28009 Madrid, Spain; sergio.gil@iisgm.com (S.G.-M.); diegocarbonell3@gmail.com (D.C.); veronica.perez@iisgm.com (V.A.P.-F.); 2Department of Hematology, Gregorio Marañón Health Research Institute (IiSGM), Gregorio Marañón University General Hospital, 28009 Madrid, Spain; 3Laboratory of Immune-Regulation, Gregorio Marañón Health Research Institute (IiSGM), Gregorio Marañón University General Hospital, 28009 Madrid, Spain; rocio.lopez@iisgm.com (R.L.-E.); rafael.correa@iisgm.com (R.C.-R.); 4Department of Clinical Microbiology and Infectious Diseases, Gregorio Marañón University General Hospital, 28009 Madrid, Spain; roberto.alonso@salud.madrid.org (R.A.); pmunoz@hggm.es (P.M.); 5School of Medicine, Complutense University of Madrid, 28040 Madrid, Spain; 6Precision Immunology Institute, Icahn School of Medicine at Mount Sinai, New York, NY 10029, USA; jochando@isciii.es; 7National Centre of Microbiology, Carlos III Health Institute, 28222 Madrid, Spain; 8Department of Occupational Risk Prevention, Gregorio Marañón University General Hospital, 28009 Madrid, Spain; ignacio.sanchezarcilla@salud.madrid.org; 9Department of Biostatistics, Gregorio Marañón Health Research Institute (IiSGM), Gregorio Marañón University General Hospital, 28009 Madrid, Spain; josemaria.bellon@salud.madrid.org

**Keywords:** COVID-19, mRNA-vaccines, humoral responses, cellular responses, SARS-CoV-2

## Abstract

Vaccination against SARS-CoV-2 has become the main method of reducing mortality and severity of COVID-19. This work aims to study the evolution of the cellular and humoral responses conferred by two mRNA vaccines after two doses against SARS-CoV-2. On days 30 and 240 after the second dose of both vaccines, the anti-S antibodies in plasma were evaluated from 82 volunteers vaccinated with BNT162b2 and 68 vaccinated with mRNA-1273. Peripheral blood was stimulated with peptides encompassing the entire SARS-CoV-2 Spike sequence. IgG Anti-S antibodies (humoral) were quantified on plasma, and inflammatory cytokines (cellular) were measured after stimulation. We observed a higher response (both humoral and cellular) with the mRNA-1273 vaccine. Stratifying by age and gender, differences between vaccines were observed, especially in women under 48 and men over 48 years old. Therefore, this work could help to set up a vaccination strategy that could be applied to confer maximum immunity.

## 1. Introduction

Since the apparition of Severe Acute Respiratory Syndrome Coronavirus 2 (SARS-CoV-2) in December 2019 [1], several vaccine candidates have been developed, highlighting a novel type of vaccine based on mRNA. These mRNA-based vaccines have proven to be a great platform for designing specific vaccines against emerging infectious diseases because they present a versatile and rapid capacity to be designed against a specific antigen [2]. This novel approach was developed years ago [3,4], but the first mRNA vaccines commercially developed were those to confer immunity against SARS-CoV-2. The first two novel vaccines were the BNT162b2 [5] and mRNA-1273 [6], designed to confer protection against the spike protein from the initial SARS-CoV-2 strain (GenBank MN908947.3). This protein is responsible for the entry into the host cells by binding to the ACE2 receptor [7], and therefore, these vaccines confer specific anti-spike-SARS-CoV-2 immunity. Therefore, the host immune system could prevent the viral infection of the coronavirus SARS-CoV-2 into the target cells by neutralising the spike protein before its binding to the ACE2 receptor [8]. Initially, both vaccines required two doses separated in time, by an interval of 21 days in the case of the BNT162b2 vaccine and 28 days in the case of the mRNA-1273. However, more recent research showed that increasing the interval between vaccine doses improved the level of IgG in plasma [9]. Following the guidelines of the administration, 30,420 and 43,548 volunteers participated in a clinical trial where they were vaccinated with the mRNA-1273 or the BNT162b2 vaccines, which showed effective protection against SARS-CoV-2. Protection was evaluated at 94.1% for the mRNA-1273 vaccine [10] and 95% for the BNT162b2 vaccine [5]. Furthermore, in these cohorts of volunteers, both vaccines were deemed to be safe, despite some local and systemic reactions.

Since the authorisation of both vaccines for their emergency use, most studies focused on humoral responses, likely due to the rapid IgG measurement in plasma. However, studying both humoral and cellular responses would broaden the current knowledge of these vaccines. Several studies have already demonstrated that these mRNA vaccines presented a fast induction of anti-spike, anti-RBD, and neutralising antibodies (humoral response) and specific T-cells (cellular response) after their administration [11,12,13,14]. These levels of antibodies and the capacity of specific T-cells to respond in contact with SARS-CoV-2 after vaccination should be enough to confer protection in the short term, reaching maximum values 2–3 weeks after the second dose [15,16,17,18]. Despite many studies showing that specific anti-SARS-CoV-2 antibodies can still be detected between 7 and 10 months after complete vaccination with the mRNA vaccines [19,20,21], little is known about the presence of the specific T-cells generated by the vaccines. Some studies showed that the production of IFN-γ, a key cytokine related to the specific anti-SARS-CoV-2 T-CD4+ memory cells in response to in vitro assays, was partially lost but still detectable over 6–8 months [22,23]. However, other cytokines implicated in the activation and response of T cells have not yet been studied.

Therefore, studying the humoral response and the specific immune cellular memory several months after vaccination with the two doses of mRNA vaccines in more depth is essential to better understand how specific anti-SARS-CoV-2 immunity evolves over months and to detect potential individuals who could need a third booster dose.

In this study, we analysed the Anti-S IgG antibodies and performed an in-depth study of inflammatory cytokines related to the cellular response 30 and 240 days after complete vaccination considering demographic variables such as age or gender.

## 2. Materials and Methods

### 2.1. Participants and Blood Extraction

We performed a prospective observational study on 150 healthy volunteers vaccinated with mRNA vaccines: BNT162b2 (*n* = 82) and mRNA-1273 (*n* = 68). The study was conducted after the approval from the University Hospital Gregorio Marañón ethics committee (MICRO.HGUGM.2020-021). Volunteers were healthcare workers from the University Hospital Gregorio Marañón of Madrid, who received both doses of the mRNA vaccines between January and February 2021. The mean days between the first and second dose was 21 days for BNT162b2 and 28 days for mRNA-1273. The mean age of the BNT162b2 volunteers was 46.52 (±1.36 years old) and 47.85 (±1.38 years old) for the mRNA-1273 volunteers. Around 80% of the volunteers were women, reflecting the population of the hospital’s healthcare staff. The rest of the cohort’s characteristics are listed in Table 1.

We collected peripheral blood samples at two time-points from the volunteers. The first time-point of recruitment of volunteers vaccinated with BNT162b2 was performed between 23 and 40 days after the second dose, with a median of 31 days, while volunteers with mRNA-1273 were recruited between 24 and 28 days, with a median of 27 days. For the volunteers vaccinated with BNT162b2, the second time-point of recruitment took place between 225 and 279 days after the second dose, with a median of 246 days (Table 1). Two volunteers were lost during the follow-up, and therefore, their second extraction samples on day 240 were not available. Another four volunteers were excluded from the second time point study since they were infected between the two dates. The volunteers vaccinated with mRNA-1273 were recruited between 227 and 266 days after receiving the second dose, with a median of 241 days. Five volunteers were lost during the follow-up, and therefore, their second extraction samples on day 240 were not available. For simplification, the first time-point was named “day 30 post-vaccination”, and the second time-point was “day 240 post-vaccination”.

### 2.2. Measurement of Specific Humoral and Cellular Responses

In whole blood, we measured SARS-CoV-2 anti-spike IgG (Anti-S IgG) by employing the Architect SARS-CoV-2 anti-S IgG on an Architect autoanalyser for immunoassays (Abbott Laboratories, Chicago, IL, USA). We also quantified the production of cytokines after the specific stimulation of whole blood with peptide pools derived from SARS-CoV-2, as published previously [24]. Briefly, fresh whole blood was directly stimulated with three different conditions: stimulation with the S-peptide pool (a combination of PepTivator SARS-CoV-2 Prot S, Prot S1 and Prot S+, Miltenyi Biotec, Bergisch Gladbach, Germany), the NM-peptide pool (combination of PepTivator SARS-CoV-2 Prot N and Prot M, Miltenyi Biotec, Bergisch Gladbach, Germany) covering the whole sequence of the nucleocapsid and membrane proteins, and a control condition without stimulation. These peptides derived from SARS-CoV-2 consisted mainly of 15 mer sequences overlapping the complete spike, nucleocapsid, and membrane regions of the SARS-CoV-2 genome (GenBank MN908947.3, Protein QHD43419.1). A total of 80 µL of RPMI with L-Glutamine (Lonza, Gampel-Bratsch, Switzerland) supplemented with antibiotics (ampicillin, cloxacillin, and gentamicin, all from Normon, Madrid, Spain) was added to 320 µL of whole blood for each condition. According to the manufacturer’s instructions, the peptide pools S and NM were used at 1 µg/mL. Whole blood and peptide pools were incubated for 14–16 h at 37 °C and 5% CO_2_. After incubation, plasma from stimulated whole blood was recovered and analysed using the microfluidic ELISA equipment ELLA-Protein Simple (Biotechne, Minneapolis, MN, USA), measuring the concentration of cytokines released after stimulation (IL-12p70, IL-10, IL-2, IL-4, IFN-γ).

For the four cytokines and the anti-S IgG, the upper and the lower limits of quantification (ULOQ and LLOQ, respectively) were calculated according to the manufacturer’s instructions. For the anti-S IgG, the ULOQ was 5680 BAU/mL, and the LLOQ was 7.1 BAU/mL. The limits of quantification for the cytokines were calculated from the limit of quantification of the ELLA equipment multiplied by 2, taking into account the dilution of the sample. For IL-10, the ULOQ was 4424 pg/mL, and the LLOQ was 1.16 pg/mL. For IL-2, the ULOQ was 4100 pg/mL, and the LLOQ was 1.08 pg/mL. For IL-4, the ULOQ was 4040 pg/mL, and the LLOQ was 1 pg/mL. Finally, for IFN-γ, the ULOQ was 8000 pg/mL, and the LLOQ was 0.34 pg/mL. In the figures, only the LLOQ of the IL-4 is presented since the values of the rest of the parameters were far from the limits.

### 2.3. Statistics

Statistical analyses were conducted at the IISGM biostatistics facility. To study the evolution of each variable, mixed-effects models adjusted by age and gender were used, taking each individual as a random effect and applying a logarithmic transformation for each variable. The mixed-effects Tobit regression was used for variables with censored data (IL-2 and IL-4). A multiple linear regression model transformed into the logarithmic of the variable was used to compare vaccines between cohorts on days 30 and 240. For IL-4, Tobit regression was used. For multiple comparisons, the Bonferroni correction method was used. To study the correlation between the humoral and cellular responses, the Spearman correlation was used. Graphics were made with GraphPad Prism Software (version 8.0, California, USA), and statistical analysis was performed with Stata Software (version 16.1, College Station, TX, USA) and IBM SPSS Statistics (version 25, Chicago, IL, USA). The figure/table legends indicated the specific tests used for each analysis. ns = *p*-value > 0.05. * *p* < 0.05. ** *p* < 0.01. *** *p* < 0.001.

## 3. Results

### 3.1. Cohort Characteristics

The sample of volunteers included in this study was composed of healthcare workers vaccinated with two doses of mRNA-1273 or BNT162b2 vaccines. There were no significant differences for all the characteristics listed in Table 1 (*p*-value > 0.05). We did not observe significant differences between cohort groups stratified by either age (< or ≥48 years old) or gender (Table 2). Four volunteers vaccinated with BNT162b2 were infected between days 30 and 240 post-vaccination and were excluded from the follow-up. The characteristics of these patients are listed in Table 3. Observing the anti-S IgG levels measured prior to infection, we observed that only patient 1 presented lower levels than the non-infected BNT162b2 volunteers. However, the measurement of the anti-S IgG for this patient was 40 days after the second vaccine dose, compared to the other three infected individuals when the measurement was taken 31 days after completing the vaccine protocol. For patients 2, 3 and 4, the anti-S IgG levels were higher than those of the non-infected vaccinated group. Therefore, focusing on these volunteers, the levels of anti-S IgG on day 30 post-vaccination could not be used to identify potential individuals more susceptible to being infected due to a poor response to vaccination.

### 3.2. Levels of Specific Anti-S Antibodies Were Higher with mRNA-1273, but Both mRNA Vaccines Presented a High Degree of Wane over the Months

The anti-S antibodies and the cytokine values were calculated with a statistical model adjusted by age and gender. We represented the estimated mean values after the model and the upper and lower confidence intervals (95% IC) for each determination on day 30 and day 240. We also used a mixed-effects model adjusted by age and gender to study the evolution of the values between both time points.

Specific anti-S IgG antibodies were measured in plasma samples from all volunteers. On day 30, we observed high levels of antibodies for both vaccines, even if mRNA-1273 volunteers presented significantly higher values than BNT162b2 (Figure 1, *p*-value = 0.001). After 8 months post-vaccination, on day 240, the levels of the antibodies had drastically waned, reaching mean values that were 10 times lower. However, mRNA-1273 recipients still presented higher values than those that had received BNT162b2 (Figure 1, *p*-value < 0.001).

Using a mixed-effects model to study the evolution of the values between day 30 and day 240, we observed that the anti-S IgG levels between both dates were significantly different for each vaccine (Figure 1, *p*-value < 0.001). We calculated the interaction between the type of vaccine and the time (V#T) with a *p*-value of 0.0331, indicating that the decline in antibody levels was significantly different between vaccines over time, the anti-S IgG antibody levels from the BNT162b2 volunteers presented a greater decrease over time.

Regarding the protective effect of such IgG levels, it was demonstrated that a level of approximately 100 BAU/mL could be protective [25]. Therefore, we calculated the number and frequency of individuals that presented a BAU/mL level inferior (considered non-protected) to 100 at day 240 post-vaccination (Table 4). We observed that 12.5% of individuals vaccinated with mRNA-1273 and 45% of individuals vaccinated with BNT162b2 presented a level of IgG that could be considered non-protective after 240 days post-vaccination.

According to these results, this huge wane in the levels of the specific anti-S antibodies could indicate that the humoral response was not mobilised again. This trend was expected since antibody levels disappear over time without further restimulation, such as a booster dose or an antigen exposure.

### 3.3. Cellular Responses to SARS-CoV-2-Derived Spike Stimulation

Not only the humoral response but also the cellular response is essential in fighting SARS-CoV-2. Therefore, we also studied cytokine production in response to the in vitro stimulation of the whole blood of individuals with the SARS-CoV-2-derived S-peptides pool. We considered volunteers with cytokine levels under the lower limit of quantification (LLOQ) to be non-responder volunteers. We observed that for the cytokines IFN-γ, IL-10, and IL-2, almost all the volunteers had a positive response (Figure 2). In the case of IL-4, we observed on day 30 that some volunteers vaccinated with BNT162b2 did not respond, while all the mRNA-1273 volunteers responded correctly. On day 240, we observed a decrease in the responding volunteers’ frequency for both vaccines. In addition, for IL-12p70, a high percentage of vaccinated individuals did not respond correctly (levels under the LLOQ), especially in BNT162b2 volunteers. Therefore, we decided to discard this cytokine’s data for further analyses and continued the analysis with IFN-γ, IL-10, IL-2, and IL-4. To sum up, BNT162b2-vaccinated individuals presented weaker IL-12p70 and IL-4 cellular responses at day 240 post-infection than mRNA-1273-vaccinated individuals.

### 3.4. Specific Cytokines Produced after the Cellular Stimulation Were Higher with mRNA-1273

We measured the production of IFN-γ due to its implication in the anti-viral Th1 response, which is necessary to fight SARS-CoV-2 [26]. In a manner similar to the antibody levels, on day 30, people vaccinated with the mRNA-1273 vaccine produced a higher amount of IFN-γ than those vaccinated with BNT162b2 (Figure 3a, *p*-value = 0.002). We also studied IL-2 and IL-4 levels, as they have been implicated in T-cell activation [27] and antibody production [28], respectively. As observed for IFN-γ production, these two cytokines’ levels were significantly higher in volunteers vaccinated with mRNA-1273 than BNT162b2 after whole-blood stimulation at day 30 post-vaccination (Figure 3b, IL-2, *p*-value < 0.001; Figure 3c, IL-4, *p*-value < 0.001).

Regarding the production of these cytokines on day 240 post-vaccination, we observed the same pattern as on day 30. A diminution of the cytokines’ production was observed for all the cytokines compared to the values observed on day 30 post-vaccination. However, we still observed a significantly higher production of IFN-γ, IL-2, and IL-4 in volunteers vaccinated with mRNA-1273 than in those vaccinated with BNT162b2 (Figure 3a, *p*-value = 0.030; Figure 3b, *p*-value = 0.001; and Figure 3c, *p*-value < 0.001).

We also studied the production of IL-10, an anti-inflammatory cytokine implicated in the balance and regulation of immune responses [29]. We observed higher cytokine levels in the mRNA-1273 volunteers compared to the BNT162b2 volunteers on day 30 (Figure 3d, *p*-value = 0.001), but not on day 240 (*p*-value = 0.093).

Therefore, the mRNA-1273 vaccine induced a significantly higher cellular-specific immunity regarding the production of cytokines after whole-blood stimulation with SARS-CoV-2-derived peptide pools compared with the BNT162b2 vaccine.

### 3.5. Anti-SARS-CoV-2 Spike-Specific Cellular Immunity Is Lost over Time Post-Vaccination

Besides comparing the cellular response of the BNT162b2 or mRNA-1273 vaccines in volunteers, we investigated whether those responses were lost over time as observed for the humoral response. We used a mixed-effects model adjusted by age and gender to study the evolution of the cellular response over time. We compared the difference between both vaccines and their interaction over time (V#T). Unlike the evolution in the humoral response, the loss of the cellular response was comparable between both groups (Figure 3a,b, *p*-value = 0.7621 for IFN-γ and *p*-value = 0.9438 for IL-2, respectively). Through this model, IL-4 production after whole-blood stimulation showed a different comportment (Figure 3c, *p*-value = 0.0234), indicating that both vaccines led to different behaviour over time. The mRNA-1273 induced a higher IL-4-associated specific cellular response than BNT162b2, but this signal presented a more pronounced decrease over time. IL-10 levels were not observed to be decreased over time. Instead, they were maintained (Figure 3d, *p*-value = 0.7893), indicating that this cytokine could be produced with the intent to control the remnant of immune activation derived from the vaccination after several months, independent of the vaccine source.

Altogether, these results indicate that the cellular immune response conferred by the mRNA-1273 vaccine is more robust than that conferred by the BNT162b2 vaccine, both after the first month post-vaccination and at day 240 post-vaccination. However, this response tended to decrease over the months for both vaccines, reaching low values on day 240 post-vaccination.

### 3.6. Correlation between Humoral and Cellular Responses

We correlated the anti-S IgG levels, and the production of the cytokines described earlier to detect differences between vaccines. Attending to the correlation between anti-S IgG and IFN-γ on day 30, we observed a positive correlation for the volunteers vaccinated with BNT162b2 (r = 0.4169, *p*-value < 0.001), but not in the case of mRNA-1273 (*p*-value = 0.162, Table 5). On day 240, the positive correlation was still present for the BNT162b2 volunteers (r = 0.2577, *p*-value = 0.026). These results indicate that the coordination between the cellular and the humoral immunity conferred by the BNT162b2 vaccine was different from that conferred by the mRNA-1273 vaccine. We also observed that both vaccines exhibited a positive correlation between anti-S IgG and IL-2 levels on day 30 (*p*-values of 0.026 and 0.016, in mRNA-1273 and BNT162b2 volunteers, respectively) but not on day 240 (*p*-values of 0.084 and 0.104, in mRNA-1273 and BNT162b2 volunteers, respectively).

After whole-blood stimulation, we studied the correlation between IFN-γ and IL-2, IL-4, or IL-10. We observed robust correlations between almost all the cytokines produced after in vitro stimulation (Table 6). According to these results, the cellular responses regarding the expression of different cytokines after stimulation are homogenous. More interestingly, the type of immunity conferred by the BNT162b2 could be different from that of the mRNA-1273 since the cellular and humoral responses did not correlate when comparing both vaccines’ responses. One can assume that the mechanisms of antigen presentation after vaccination could differ between vaccines and thus could explain that the humoral and specific Th1 cellular responses were correlated in the BNT162b2 volunteers but not in the mRNA-1273 volunteers.

### 3.7. The Implication of Age and Gender on the Immune Response Achieved after Vaccination

To identify potential cohort groups in which vaccination could perform poorly, we stratified volunteers by two different factors, age and gender. To stratify individuals by age, we calculated the median age of the vaccinated cohort, which was 48 years old. This value was in line with the age-related regression of the thymus, which showed a decline in naïve T cell output in individuals around 40–50 years, and, thus, could compromise the generation of a new robust anti-SARS-CoV-2 memory cell [30]. Therefore, we divided the cohort into volunteers younger than 48 years old and older than 48 years old (≥48 years old). We detail the characteristics of the volunteers of every cohort group in Table 2.

In regard to the anti-S IgG levels, we observed on days 30 and 240 that women older than 48 years vaccinated with mRNA-1273 had higher levels than the same cohort vaccinated with BNT162b2 (Figure 4a, *p*-value = 0.045, and Figure 4b, *p*-value = 0.002). Even though we did not observe differences in the other cohort groups, those vaccinated with mRNA-1273 had a higher global response.

Regarding the production of cytokines after whole-blood stimulation with the SARS-CoV-2-derived spike peptide pool, we observed several significant differences on day 30 and day 240 post-vaccination between stratified groups. Globally, almost all stratified groups showed higher levels of specific anti-S cellular responses when individuals were vaccinated with mRNA-1273 than those vaccinated with BNT162b2. Early after vaccination (day 30), the IFN-γ, IL-2, and IL-4 responses were significantly higher in women younger than 48 years old (Figure 5a, *p*-value = 0.042; Figure 5c, *p*-value < 0.001, and Figure 5e, *p*-value < 0.001, respectively). This pattern was also observed in women over 48 years old for the IL-4 and the IL-10-specific responses (Figure 5e, *p*-value < 0.001, and Figure 5g, *p*-value = 0.006, respectively). Thus, women presented significantly higher specific cellular responses, independent of their age, when vaccinated with mRNA-1273 than with BNT162b2.

Months after vaccination (day 240), men older than 48 years old presented higher specific cellular responses when vaccinated with mRNA-1273 compared with those vaccinated with BNT162b2 (Figure 5b, *p*-value = 0.025; Figure 5d, *p*-value = 0.001; Figure 5f, *p*-value < 0.001; and Figure 5h, *p*-value = 0.048, respectively).

Altogether, these results indicate that mRNA-1273 induced a more robust cellular response in almost all cohort groups at both sampling times post-vaccination. Women under 48 years old vaccinated with mRNA-1273 presented higher immunity, especially shortly after vaccination. In contrast, men over 48 years old vaccinated with mRNA-1273 seemed to have a higher immunity after 240 days post-vaccination.

## 4. Discussion

The development of mRNA-based vaccines has led to a new paradigm in vaccine development. Since these vaccines were distributed globally, they have demonstrated good effectiveness, reducing infection, severity, hospitalisation, and mortality in cases of COVID-19 infection [31,32]. However, many studies have described differences between vaccines, with the mRNA-1273 vaccine inducing a higher production of antibodies against SARS-CoV-2 than the BNT162b2, with these differences being maintained over months [21,33,34,35,36]. A few groups have already studied the cellular response in terms of the production of IFN-γ after specific SARS-CoV-2 in vitro stimulation and observed that, like the antibody levels, the production of this cytokine was maintained over months [22,23]. Moreover, they also observed that the mRNA-1273 vaccine induced an immunity capable of producing higher levels of IFN-γ compared to BNT162b2. However, as far as we know, no study exists comparing both humoral and cellular responses, including not only cytokines implicated in the Th1 response (such as IFN-γ), but also others such as IL-2 (Th1 response, reinforcing the results concerning IFN-γ), IL-4 (Th2 response), and IL-10 (immune-regulatory response), between mRNA vaccines with a follow-up of 240 days post-vaccination. We observed that both antibody levels and T-cell responses were superior in volunteers vaccinated with the mRNA-1273 vaccine after 30 and 240 days post-vaccination compared to those vaccinated with BNT162b2. These results could explain why in our cohort, four volunteers vaccinated with the BNT162b2 vaccine were infected after vaccination, while those vaccinated with mRNA-1273 remained uninfected. This greater drop in both humoral and cellular response in BNT162b2 volunteers could be related to the effectiveness of both vaccines in real-world administration. Indeed, months after the administration, it was observed that the mRNA-1273 vaccine presented higher effectiveness than the BNT162b2 vaccine whenever vaccine effectiveness was compared [32,37,38]. We previously reported that the humoral and cellular responses on the first 14 days post-vaccination reached a maximum peak on day 14 [15], which we observed to be maintained on day 30. However, as many others have already observed, this response is partially reduced on day 240, showing an evident loss of the humoral and cellular response over time. These results correlate with that already observed by Puranik et al., that the effectiveness of the vaccines is high in the first weeks post-vaccination but is partially reduced in the following months [32]. More recently, in a cohort of 222,493 individuals, it was determined that at least 67% of individuals were protected for 5–8 months after two BNT162b2 doses [25]. Regarding those results, the authors determined that IgG level protection was >94 BAU/mL for BNT162b2, >107 BAU/mL for the ChAdOx1 vaccine, and >33 BAU/mL for natural infection. Unfortunately, no level was quantified as protective for the mRNA-1273 vaccine, but regarding the previous results [25], one can assume that protection could be reached at around 100 BAU/mL. If we consider the level of protection at 100 BAU/mL, on day 240 post-vaccination, 33 of 74 individuals vaccinated with BNT162b2 (45%) and 7 of 56 (12.5%) vaccinated with mRNA-1273 presented less than 100 BAU/mL and therefore, could be considered non-protected. No protection level was determined for cellular immunity.

Since there were no demographic differences between both cohorts, these differences observed in the humoral and the cellular response could be due to the differences between the vaccines. Apart from an in vitro study by Aldén et al. [39], which revealed that the LINE-1 retrotransposon was implicated in the reverse transcription of the BNT162b2 vaccine in a human liver cell line, there are no studies focused on the cellular mechanisms implicated during vaccination that could explain the differences between mRNA-1273 and BTN162b2. Here, we observed a strong correlation between anti-S IgG levels and IFN-γ for the BNT162b2 vaccine, but not for mRNA-1273. Moreover, the evolution of the response in the anti-S IgG and the IL-4 levels is different between vaccines. Both vaccines are mRNA-based and are designed to confer immunity against the spike protein of SARS-CoV-2. They are based on lipid nanoparticles containing mRNA that enter the host cell, where the spike protein of SARS-CoV-2 is synthesised and leads to the induction of humoral and cellular immunity [40,41]. However, the composition of the vaccines differs. The BNT162b2 vaccine is composed of the mRNA sequence, ALC-0315, ALC-0159, DSPC, and cholesterol, while the mRNA-1273 is composed of the mRNA sequence, SM-102, PEG-DMG, DSCP, and cholesterol [42]. Furthermore, although the administration is intramuscular for both vaccines (deltoid muscle), the dosage per injection in the case of BNT162b2 is 0.3 mL, containing 30 µg of mRNA per injection (concentration of 100 µg/mL), separated by 21 days [43], while for mRNA-1273, each injection is 0.5 mL, containing 100 µg of mRNA (concentration of 200 µg/mL) per injection, separated by 28 days [44]. Therefore, the differences in the composition and concentration could explain the differences observed between vaccines in terms of effectiveness and the humoral and cellular responses. Moreover, the difference in the time of administration (21 vs. 28 days) could also explain these differences since it is known that a delayed interval in the administration of the doses of the BNT162b2 vaccine achieves a better response than the 21-day administration schedule [45]. Although both vaccines are based on the same lipid nanoparticles concept, each vaccine could trigger different cellular mechanisms after administration. More in-depth studies are required to better understand all the cellular mechanisms implicated in the conferring of specific anti-SARS-CoV-2 immunity.

Age has already been described as a potential risk factor in COVID-19 severity [46,47,48]. Moreover, it has already been seen that vaccinated older adults present lower levels of antibodies in comparison with younger adults [21], and consequently, they present lower vaccine effectiveness and a higher risk of infection or re-infection [49,50,51]. Indeed, in our work, we observed a major wane in the humoral and cellular responses in older adults vaccinated with BNT162b2 than those vaccinated with mRNA-1273. Altogether, we could establish a preferred vaccination schedule, depending on age and gender. For women of all ages, vaccination with the mRNA-1273 vaccine would be recommended since it confers higher levels of anti-S IgG in comparison with BNT162b2 at both 30 and 240 days post-vaccination. Regarding the cellular response, the mRNA-1273 vaccine also confers a higher production capacity of cytokines related to the Th1, Th2, and immune-regulatory responses, especially on day 30 and day 240 post-vaccination. In addition, all four of the infected individuals post-vaccination were women vaccinated with BNT162b2, which could indicate that mRNA-1273 not only confers a better immunity but also provides better protection against infection for women. Of course, further study must be made to confirm such a conclusion since the number of infected individuals was low in the studied cohort.

For the male cohort, we could not determine if one vaccine was better than the other, partly because of the reduced number of individuals in each group (14 vaccinated with mRNA-1273 versus 8 vaccinated with BNT162b2). Other studies showed that older age, male sex, and long-term health conditions were all associated with substantially low peak levels in participants who received BNT162b2 [25,50]. However, increasing the number of patients in the present cohort could provide a more robust statistical analysis to determine possible differences in the cellular immune response between vaccines.

Some of the factors considered in this work could be extrapolated and measured in other body fluids, for example, saliva or nasal mucosa, and would help to better determine the state of patients attending medical offices. As previously described [52,53], SARS-CoV-2 or antibodies against coronavirus could be detected in several body fluids, and this could overcome some of the limitations presented by the determination of antibodies in plasma, which requires blood extraction. However, Azzi et al. [52] described a poor correlation between serum and saliva levels regarding specific IgG and IgA, observing reduced levels in saliva but not in serum. Therefore, the fast detection of specific cellular immunity in this type of sample is not feasible since it requires the presence of peripheral blood mononuclear cells (PBMCs) and the in vitro stimulation of those cells. Therefore, although the detection of the humoral response in saliva or nasal mucosa is useful for the quick detection of anti-SARS-CoV-2 antibodies, the in vitro stimulation of PBMCs is still the main method of detecting cellular memory in volunteers.

The limitations of the study are firstly based on the number of recruited participants. The reduced number of male individuals did not allow us to obtain robust statistics; further studies are therefore needed. Moreover, we studied only two of all the currently available vaccines since these two vaccines were the only two accepted at the time for the vaccination of healthcare workers. Moreover, the present follow-up could be challenging to continue since these vaccines were developed against the original SARS-CoV-2 variant, and it has been shown that these vaccines could be less efficient against other variants such as SARS-CoV-2 Omicron, for example [54,55].

In conclusion, we found that the mRNA-1273 vaccine confers better humoral and cellular responses than BNT162b2, both on days 30 and 240 post-vaccination. This could indicate that the mRNA-1273 vaccine is more protective against SARS-CoV-2 infection than the BNT162b2 vaccine. Continuing follow-up would allow us to observe the evolution of these responses, the influence of the third mRNA dose, and, more importantly, their influence in terms of protection against potential newly emerging variants.

## Figures and Tables

**Figure 1 biomedicines-10-01676-f001:**
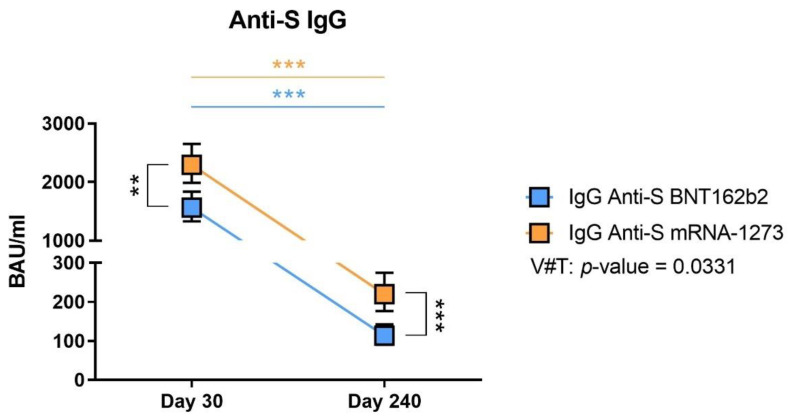
Anti-S IgG levels on day 30 and 240 post-vaccination and predicted evolution of the values. The estimated means and confidence intervals were calculated by a multiple linear regression model, adjusted by age and gender and with Bonferroni correction. Squares represent the mean and confidence intervals for anti-S IgG BAU/mL at days 30 and 240, for BNT162b2 (in blue) and mRNA-1273 (in orange). The evolution of anti-S IgG values was calculated using the mixed-effects model. V#T indicates the interaction between the type of vaccine and the time evolution. ** *p*-value < 0.01. *** *p*-value < 0.001.

**Figure 2 biomedicines-10-01676-f002:**
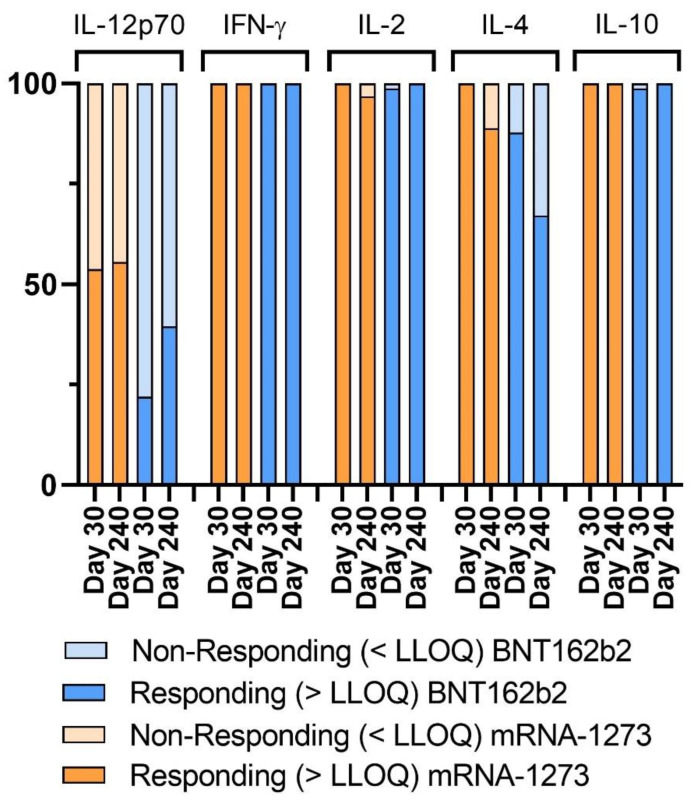
Percentage of responding volunteers reaching the lower limit of detection (LLOD). Bar plot representing the percentage of responding volunteers vaccinated with BNT162b2 (orange) or mRNA-1273 (blue) or non-responding (<LLOD, light orange and light blue, respectively) for the cytokines IL-12p70, IFN-γ, IL-2, IL-4, and IL-10.

**Figure 3 biomedicines-10-01676-f003:**
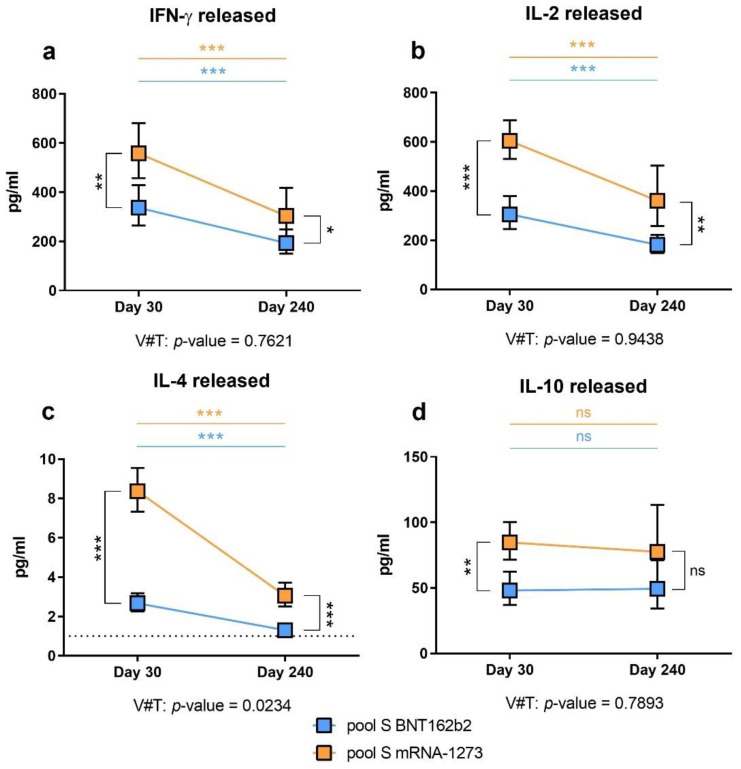
Specific cellular immunity after stimulation with the S pool. The estimated means and confidence intervals were calculated by multiple linear regression model, adjusted by age and gender and with Bonferroni’s correction. Squares represent the mean and confidence intervals for IFN-γ (**a**), IL-2 (**b**), IL-4 (**c**), and IL-10 (**d**), at day 30 and 240, for BNT162b2 (in blue) and mRNA-1273 (in orange). Lower limit of detection (LLOD) is indicated by the dotted line for IL-4 (1 pg/mL). The evolution for each cytokine was assessed by the mixed-effects model. V#T indicates the interaction between the type of vaccine and the time evolution. Non-significant (ns = *p*-value > 0.05). * *p*-value < 0.05. ** *p*-value < 0.01. *** *p*-value < 0.001.

**Figure 4 biomedicines-10-01676-f004:**
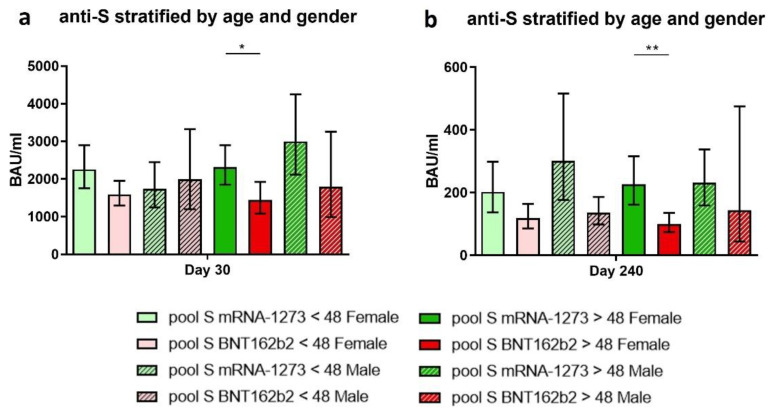
Anti-S IgG at days 30 and 240 stratified by age and gender. The estimated means and confidence intervals were calculated by a multiple linear regression model stratified by age and gender. Bonferroni’s correction was used. Bars represent the mean and confidence intervals for anti-S IgG BAU/mL at days 30 (**a**) and 240 (**b**). Each graph represents volunteers vaccinated with mRNA-1273: women < 48 years (light green), women > 48 years (dark green), men < 48 years (striped light green), and men > 48 years (striped dark green). Volunteers vaccinated with BNT162b2: women < 48 years (light red), women > 48 years (dark red), men < 48 years (striped light red), and men > 48 years (striped dark red). * *p*-value < 0.05. ** *p*-value < 0.01.

**Figure 5 biomedicines-10-01676-f005:**
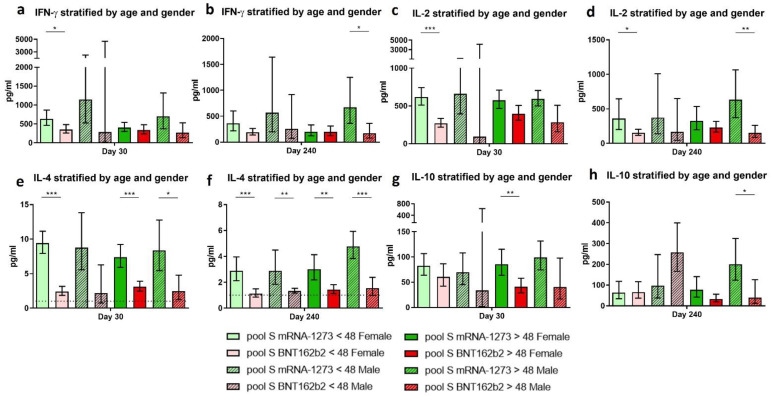
Anti-SARS-CoV-2 spike specific cellular response regarding the individuals’ age and gender. The estimated means and confidence intervals were calculated by a multiple linear regression model, stratified by age and gender. Bonferroni’s correction was used. Bars represent the mean and confidence intervals for IFN-γ (**a**,**b**), IL-2 (**c**,**d**), IL-4 (**e**,**f**), and IL-10 (**g**,**h**), at day 30 and 240, respectively. Each graph represents volunteers vaccinated with mRNA-1273: women < 48 years (light green), women > 48 years (dark green), men < 48 years (striped light green), and men > 48 years (striped dark green). Volunteers vaccinated with BNT162b2: women < 48 years (light red), women > 48 years (dark red), men < 48 years (striped light red), and men > 48 years (striped dark red). Lower limit of detection (LLOD) is indicated by the dotted line for IL-4 (1 pg/mL). * *p*-value < 0.05. ** *p*-value < 0.01. *** *p*-value < 0.001.

**Table 1 biomedicines-10-01676-t001:** Characteristics of the vaccinated cohort.

Characteristics	mRNA-1273	BNT162b2	*p*-Value
**Volunteers**, *n*	68	82	
**Age** (years), mean (SEM)	47.85 (± 1.38)	46.52 (± 1.36)	0.495
**Days between 2nd dose and first extraction**, median (range)	27 (24–28)	31 (23–40)	
**Days between 2nd dose and second extraction**, median (range)	241 (227–266)	246 (225–279)	
**Gender**, *n* (%)			0.275
Male	14 (20.6)	11 (13.4)	
Female	54 (79.4)	71 (86.6)	
**Ethnicity**, *n* (%)			1.000
Caucasian	65 (95.6)	78 (95.1)	
Latin-American	3 (4.4)	4 (4.9)	
**Comorbidities**, *n* (%)			
Hyperthyroidism	0 (0.0)	1 (1.2)	0.469
Hypothyroidism	5 (7.4)	1 (1.2)	0.091
Hypertension	5 (7.4)	3 (3.6)	1.000
Dyslipidemia	3 (4.4)	4 (4.8)	1.000
Hypercholesterolemia	0 (0.0)	2 (2.4)	0.501
Diabetes	0 (0.0)	1 (1.2)	1.000
Pulmonary disease	5 (7.4)	2 (2.4)	0.245

Characteristics of the cohort vaccinated with mRNA-1273 or BNT162b2 on days 30 and 240. Listed are the number of patients for each vaccine and characteristics such as age (expressed as mean ± SEM), gender (expressed as number of patients with each characteristic, and in asterisks, the percentage over the total number of patients for each cohort), ethnicity, and comorbidities. Also listed are the median number of days between the first and second dose and the days of blood sample extraction. The two-sample *t*-test was used for the comparison of age, and Fisher’s exact test for the rest of the characteristics. *p*-value < 0.05 indicates statistical difference between groups. SEM, standard error of the mean.

**Table 2 biomedicines-10-01676-t002:** Characteristics of the vaccinated cohort, stratified by age and gender.

Characteristics	mRNA-1273 < 48 Female	BNT162b2 < 48 Female	mRNA-1273 < 48 Male	BNT162b2 < 48 Male	mRNA-1273 > 48 Female	BNT162b2 > 48 Female	mRNA-1273 > 48 Male	BNT162b2 > 48 Male
**Volunteers**, *n*	26	37	6	3	28	34	8	8
**Ethnicity**, *n* (%)								
Caucasian	24 (92.3)	35 (94.6)	6	3	27 (96.4)	32 (94.1)	8	8
Latin-American	2 (7.7)	2 (5.4)	0	0	1 (3.6)	2 (5.9)	0	0
**Comorbidities**, *n* (%)								
Hyperthyroidism	0	1 (2.7)	0	0	0	0	0	0
Hypothyroidism	2 (7.7)	0	0	0	3 (10.7)	1 (2.9)	0	0
Hypertension	0	0	0	1 (33.3)	4 (14.3)	2 (5.9)	1 (12.5)	0
Dyslipidemia	0	0	0	0	2 (7.1)	3 (8.8)	1 (12.5)	1 (12.5)
Hypercholesterolemia	0	1 (2.7)	0	0	0	1 (2.9)	0	0
Diabetes	0	0	0	0	0	1 (2.9)	0	0
Pulmonary disease	2 (7.7)	1 (2.7)	0	0	2 (7.1)	1 (2.9)	1 (12.5)	0

Characteristics of the cohort vaccinated with mRNA-1273 or BNT162b2, stratified by age and gender. We calculated the median age to be 48 years. We decided to divide the volunteers into several sub-groups: mRNA-1273 < 48 female, BNT162b2 < 48 female, mRNA-1273 < 48 male, BNT162b2 < 48 male, mRNA-1273 > 48 female, BNT162b2 > 48 female, mRNA-1273 > 48 male and BNT162b2 > 48 male. Listed are the number of patients for each group and characteristics such as ethnicity and comorbidities.

**Table 3 biomedicines-10-01676-t003:** Characteristics of the patients infected between day 30 and day 240 post-vaccination.

Patient	Gender	Age Years	Anti-S IgG (BAU/mL)	Days between 2nd Dose and Positive Test	Days between 2nd Dose and Anti-S IgG Measurement	Days between Anti-S IgG Measurement and Positive Test	Anti-SARS-CoV-2 Test (Variant)
Patient 1	Female	63	964.24	66	40	26	PCR test (B.1.1.7)
Patient 2	Female	62	3909.64	170	31	139	PCR test
Patient 3	Female	26	2121.22	183	31	152	Antigen test
Patient 4	Female	54	4332.11	185	31	154	PCR test (B.1.617)
Rest of BNT162b2 volunteers	Male *n* = 11 Female *n* = 67	46 [22–65]	1939.42	-	32	-	-

For each patient are listed the gender, age, Anti-S IgG levels and days concerning the vaccination, IgG measurement and SARS-CoV-2 testing. We calculated the mean age and anti-S IgG for the rest of the volunteers vaccinated with BNT162b2.

**Table 4 biomedicines-10-01676-t004:** Number and frequency of individuals presenting an IgG level superior or inferior to 100 BAU/mL at day 240 post-vaccination.

	Total Non-Protected at Day 240 Post-Vaccination N (%)
Number of volunteers vaccinated with **mRNA-1273** with IgG (<100 BAU/mL // >100 BAU/mL)	7 // 56 **(12.5%)**
Number of volunteers vaccinated with **BNT162b2** with IgG (<100 BAU/mL // >100 BAU/mL)	33 // 74 **(45%)**

**Table 5 biomedicines-10-01676-t005:** Correlation between humoral and cellular responses on days 30 and 240.

**Day 30 Post-Vaccination**	**r for mRNA-1273**	***p*-Value**	**r for BNT162b2**	***p*-Value**
Anti-S IgG–IFN-γ	0.1728	0.162	0.4169	**<0.001**
Anti-S IgG–IL-2	0.2717	**0.026**	0.2659	**0.016**
Anti-S IgG–IL-4	0.0390	0.753	0.0304	0.799
Anti-S IgG–IL-10	0.2111	0.086	0.0974	0.386
**Day 240 Post-Vaccination**	**r for mRNA-1273**	***p*-Value**	**r for BNT162b2**	***p*-Value**
Anti-S IgG–IFN-γ	0.1904	0.159	0.2577	**0.026**
Anti-S IgG–IL-2	0.2367	0.084	0.1902	0.104
Anti-S IgG–IL-4	0.1882	0.177	0.1025	0.478
Anti-S IgG–IL-10	0.0073	0.957	0.1691	0.149

We performed Spearman correlations between humoral (anti-S IgG levels) and cellular responses. R value of the Spearman correlation is listed for mRNA-1273 and BNT162b2 volunteers. *p*-value of the correlation is also listed. Significant differences are highlighted in bold.

**Table 6 biomedicines-10-01676-t006:** Correlation between cellular responses on days 30 and 240.

**Day 30 Post-Vaccination**	**r for mRNA-1273**	***p*-Value**	**r for BNT162b2**	***p*-Value**
IFN-γ–IL-2	0.4041	**<0.001**	0.7708	**<0.001**
IFN-γ–IL-4	0.1683	0.173	0.5305	**<0.001**
IFN-γ–IL-10	0.3577	**0.003**	0.4533	**<0.001**
**Day 240 Post-Vaccination**	**r for mRNA-1273**	***p*-Value**	**r for BNT162b2**	***p*-Value**
IFN-γ–IL-2	0.6332	**<0.001**	0.8145	**<0.001**
IFN-γ–IL-4	0.5723	**<0.001**	0.5442	**<0.001**
IFN-γ–IL-10	0.3486	**0.008**	0.4464	**<0.001**

We performed Spearman correlations for the cellular responses. R value of the Spearman correlation is listed for mRNA-1273 and BNT162b2 volunteers. *p*-value of the correlation is also listed. Significant differences are highlighted in bold.

## Data Availability

Data is contained within the article.

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
