# Peer review of "Cellular and Humoral Responses Follow-up for 8 Months after Vaccination with mRNA-Based Anti-SARS-CoV-2 Vaccines"

_biomedicines, 2022, doi:10.3390/biomedicines10071676_

Round 1

Reviewer 1 Report

The article shows some intereting aspects especially the tentative to have a clear study on long term cellular and humoral effects after popular mRNA-based vaccine administration(s).

Here are my criticisms:

Many important references are not present for convenience. I recommend to the Authors to cite in the Introduction and Discussion other studies that in some way reduced not all but many novelties.

Why Authors divide the population enrolled (age 48) if the clusters are not significative in numbers. Tha Table S1 should be added in the main text and not in Supplementary. Moreover, Table S2 also should be included.

If readers would like to know more about IL-12p70 introduced in Figure 2 how they find out more in other Figures and Results; what Authors think about day 30 day 240 columns of IL-12 BNT162b2 in Figure 2.

Asterisks or ns in Figure 3 panel d, IL-10 day 240.

The standard deviations in the histograms of section 3.7 Figure 4 and more prominent in Figure 5 make it difficult to discuss clean and accurate results. Some error bars really make the job questionable.

More than half of the discussion could be improved. The final question is the levels that You find at day 30 and more important at day 240 are suffcient to confer immunity and are comparable with waht is expeted from published data. The sentence, "we found that the mRNA-1273 vaccine confers better humoral and cellular responses than BNT162b2, both on days 30 and 240 post-vaccination", means what for subjects vaccinated that are reading Your manuscript.

Author Response

Reviewer 1

The article shows some interesting aspects especially the tentative to have a clear study on long term cellular and humoral effects after popular mRNA-based vaccine administration(s).

Here are my criticisms:

Many important references are not present for convenience. I recommend to the Authors to cite in the Introduction and Discussion other studies that in some way reduced not all but many novelties.

Response: As suggested by the reviewer, we modified and added new comments and references to actualize the Introduction and Discussion.

Why Authors divide the population enrolled (age 48) if the clusters are not significant in numbers. Table S1 should be added in the main text and not in Supplementary. Moreover, Table S2 also should be included.

Response: We thank the reviewer for the comments. We decided to divide our cohort population by age 48 as a cut-off for several reasons. First of all, as previously described by Bajaj et al. (https://doi.org/10.3389/fphys.2020.571416), around 40-50 years of age, the remainder of the thymus further degenerates, reducing the production of naïve CD4+ T cells by the thymus by more than 99% compared with the capacity of the thymus of a newborn. The fact that in the periphery of volunteers older than 50 years old, the naïve CD4+ T cells were drastically reduced, could compromise the generation of a new robust immune response. Additionally, in our study, the median age was 48 years old. Taking these facts together, we decided to divide the study population by the threshold of 48 years old. We include a paragraph in the section “3.7 The implication of age and gender on the immune response achieved after vaccination” explaining the criteria for selecting 48 years old as the cut-off point. Following the comment, we have also included Table S1 and Table S2 in the main text as Table 3 and Table 4, respectively.

If readers would like to know more about IL-12p70 introduced in Figure 2 how they find out more in other Figures and Results; what Authors think about day 30 day 240 columns of IL-12 BNT162b2 in Figure 2.

Response: We decided not to include the IL12-p70 results because more than 50% of the values were below the limit of detection (LOD). Moreover, for those that presented detectable IL-12p70 levels, levels were very low and close to the LOD (=1.24 pg/ml) with a mean of 3.84 pg/ml and 2.65 pg/ml for the BNT162b2 vaccine on days 30 and 240, respectively, and 4.08 pg/ml and 2.26 pg/ml for the mRNA-1273 on days 30 and 240, respectively. Therefore, with only 50% of the cohort volunteers presenting very low levels of IL-12p70, we thought the conclusions would not be robust enough to be presented in the manuscript.

Asterisks or ns in Figure 3 panel d, IL-10 day 240.

Response: As suggested by the reviewer, we include ns in Figure 2 panel d, IL-10 day 240, as well as between day 30 and 240 in mRNA-1273 and BNT162b2 levels.

The standard deviations in the histograms of section 3.7 Figure 4 and more prominent in Figure 5 make it difficult to discuss clean and accurate results. Some error bars really make the job questionable.

Response: Your comment, along with the comment of the other reviewer, is related to the small number of individuals in some subgroups, especially those concerning men. This is because we decided to divide our cohort into several groups, and some of them contained a reduced number of individuals and therefore presented a higher variability. We modified the main text to avoid giving robust conclusions regarding these volunteer groups.

More than half of the discussion could be improved. The final question is the levels that You find at day 30 and more important at day 240 are suffcient to confer immunity and are comparable with waht is expeted from published data. The sentence, "we found that the mRNA-1273 vaccine confers better humoral and cellular responses than BNT162b2, both on days 30 and 240 post-vaccination", means what for subjects vaccinated that are reading Your manuscript.

Response: We appreciate this important comment, and we have modified the Discussion to improve it and add some recent works about the protection conferred by the vaccines regarding the IgG level in serum. We have also added a new Table 4, which shows the frequencies of the non-protected individuals at day 240 post-vaccination regarding the vaccine. This information came from the publication: “Wei, J.; Pouwels, K.B.; Stoesser, N.; Matthews, P.C.; Diamond, I.; Studley, R.; Rourke, E.; Cook, D.; Bell, J.I.; Newton, J.N.; et al. Antibody responses and correlates of protection in the general population after two doses of the ChAdOx1 or BNT162b2 vaccines. Nat Med 2022, 28, 1072-1082, doi:10.1038/s41591-022-01721-6.” (added as a reference), where the authors showed that the IgG level of protection is around 100 BAU/ml and is maintained for 8 months. Therefore, thanks to these data, we could describe almost 45% of the BNT162b2 individuals and 12.5% of the mRNA-1273 individuals as no longer protected at 8 months post-vaccination.

Reviewer 2 Report

-          Language and syntax needs native editing

-          Was there any sample size calculation to ensure the power of study?

-          The number of male patients was very limited. Therefore, I suggest avoiding gender based remarks and conclusions

-          Study of the presence of cellular and humoral immunity in body fluids eg. Saliva and the correlation between blood and saliva is important and has to be investigated. If present point of care testing can be utilized in critical settings like dental and medical offices.  Expand this as a suggestion/limitation and cite  below and other similar articles:

“Testing for COVID-19 in dental offices: Mechanism of action, application, and interpretation of laboratory and point-of-care screening tests. The Journal of the American Dental Association 152 (7), 514-525. e8 “

“Characteristics and detection rate of SARS-CoV-2 in alternative sites and specimens pertaining to dental practice: an evidence summary. Journal of Clinical Medicine 10 (6), 1158”

Author Response

Reviewer 2

Language and syntax needs native editing

Response: We send the manuscript to the MDPI English edition service.

-          Was there any sample size calculation to ensure the power of study?

Response: Unfortunately, the sample size was not calculated during the design phase of the study, since as we have detailed in the methodology section, we decided to model our data fitting mixed-effects regression models. The state-of-the-art for sample size estimation in mixed-effects regression is done through a simulation from available data. At the time of the development of this study during the SARS-CoV-2 pandemic and due to the novelty of our approach, there was simply no available literature that would allow us to appropriately specify the main effects, the interaction terms, and the mixed effects of the regression model. The only alternative available at the time was to make strong a priori assumptions about the model parameters and generate artificial data, which, due to the exploratory and observational nature of the study, we dismissed as inappropriate.

-          The number of male patients was very limited. Therefore, I suggest avoiding gender based remarks and conclusions

Response: As suggested by the reviewer, we modified the Discussion section to avoid giving robust conclusions regarding the male cohort.

-          Study of the presence of cellular and humoral immunity in body fluids eg. Saliva and the correlation between blood and saliva is important and has to be investigated. If present point of care testing can be utilized in critical settings like dental and medical offices. Expand this as a suggestion/limitation and cite below and other similar articles:

“Testing for COVID-19 in dental offices: Mechanism of action, application, and interpretation of laboratory and point-of-care screening tests. The Journal of the American Dental Association 152 (7), 514-525. e8 “

“Characteristics and detection rate of SARS-CoV-2 in alternative sites and specimens pertaining to dental practice: an evidence summary. Journal of Clinical Medicine 10 (6), 1158”

Response: We include the cited articles and modify the Discussion section to integrate the reviewer’s comment in our main text.

Round 2

Reviewer 1 Report

The Article is improved in the right way, more comments in the sections and extensively renewed with the sentences the two Tables and the tentative to present a better rationale study.

Except one answer to my criticism, I think that the manuscript is good.

Reviewer 2 Report

all comments are answered and the manuscript is improved